# CYCLO 🚲: Cyclic Graph Transformer Approach to Multi-Object Relationship Modeling in Aerial Videos

**Trong-Thuan Nguyen[1], Pha Nguyen[1], Xin Li[2], Jackson Cothren[1], Alper Yilmaz[3], Khoa Luu[1]**

[1]University of Arkansas    [2]State University of New York at Albany    [3]Ohio State University

[1]{thuann, panguyen, jcothre, khoaluu}@uark.edu    [2]xli48@albany.edu    [3]yilmaz.15@osu.edu

uark-cviu.github.io/projects/CYCLO

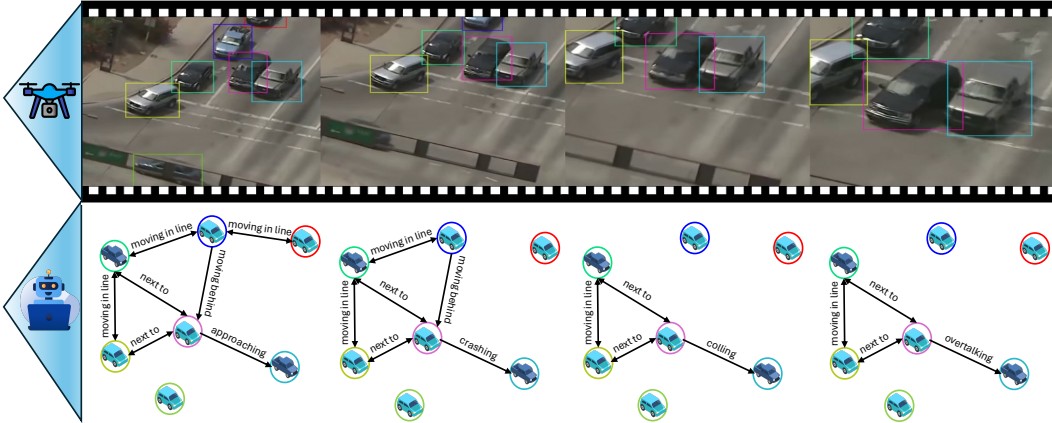

Figure 1: Multi-Object Relationship Modeling in Aerial videos analyzes a drone-captured video to detect and refine object relationships over time. The CYCLO model first identifies relationships between objects in individual frames and then incorporates temporal information about object positions and interactions to refine the understanding of those relationships across the video sequence. (Best viewed in colors)

## Abstract

Video scene graph generation (VidSGG) has emerged as a transformative approach to capturing and interpreting the intricate relationships among objects and their temporal dynamics in video sequences. In this paper, we introduce the new Aero-Eye dataset that focuses on multi-object relationship modeling in aerial videos. Our AeroEye dataset features various drone scenes and includes a visually comprehensive and precise collection of predicates that capture the intricate relationships and spatial arrangements among objects. To this end, we propose the novel Cyclic Graph Transformer (CYCLO) approach that allows the model to capture both direct and long-range temporal dependencies by continuously updating the history of interactions in a circular manner. The proposed approach also allows one to handle sequences with inherent cyclical patterns and process object relationships in the correct sequential order. Therefore, it can effectively capture periodic and overlapping relationships while minimizing information loss. The extensive experiments on the AeroEye dataset demonstrate the effectiveness of the proposed CYCLO model, demonstrating its potential to perform scene understanding on drone videos. Finally, the CYCLO method consistently achieves State-of-the-Art (SOTA) results on two in-the-wild scene graph generation benchmarks, *i.e.*, PVSG and ASPIRe.

38th Conference on Neural Information Processing Systems (NeurIPS 2024).

# 1 Introduction

Visual scene understanding has shown significant progress in extracting semantic information from images and videos using deep learning algorithms [1, 2, 3]. Building upon the significant progress in visual scene understanding using deep learning algorithms [4, 5], video scene graph generation (VidSGG) extends the concept of Scene Graph Generation (SGG) from static images to dynamic video, representing object relationships within a graph structure that evolves over time. VidSGG [6, 7, 8] focus on the temporal dimension by constructing a dynamic graph structure, encapsulating the spatial and temporal relationships among object interactions across frames. This helps in understanding human-object interactions [9, 10], temporal events [11, 12, 13], and reasoning [14, 15]. However, drone-captured videos present unique challenges due to larger image sizes and higher object density in Unmanned Aerial Vehicle (UAV) datasets [16, 17, 18]. Despite recent advances in tiny object detection [19, 20, 21], current algorithms still need to effectively model object interactions and their temporal evolution in aerial videos, which have various applications in surveillance, disaster response.

In this paper, we introduce AeroEye, the first dataset for Video Scene Graph Generation in drone-captured videos featuring **Aer**ial-**O**blique-**Eye** views. AeroEye distinguishes itself by showcasing a rich tapestry of aerial videos and an extensive set of predicates describing the intricate relations and positions of multi-objects. To address multi-object relationship modeling in aerial videos from the AeroEye dataset, we propose the Cyclic Graph Transformer (CYCLO). This new approach can establish circular connectivity among frames and enables the model to capture direct and long-range temporal relationships. By continuously updating history across a ring topology, CYCLO allows the model to handle sequences with inherent cyclic patterns, facilitating the processing of object relationships in the correct temporal order. Furthermore, CYCLO provides several advantages to VidSGG, including the ability to model periodic and overlapping relationships, predict object interactions by reasoning from previous cycles, facilitate information transfer across frames, and efficiently utilize long sequences, addressing the limitations of prior methods [22, 8]. They usually struggle with long-term dependencies due to the diminishing influence of inputs over time.

**Contributions of this Work.** There are three main contributions to this work. First, we introduce a new *AeroEye* dataset for VidSGG in drone videos, augmented with numerous predicates and diverse scenes to capture the complex relationships in aerial videos. Second, we propose the CYCLO approach, utilizing circular connectivity among frames to enable periodic and overlapping relationships. It allows the model to capture long-range dependencies and process object interactions in the appropriate temporal arrangement. Finally, the proposed CYCLO approach outperforms prior methods on two large-scale in-the-wild VidSGG datasets, including PVSG [7] and ASPIRe [8]. Interestingly, using the same method (*e.g.*, our CYCLO), the ratio of correct predictions to incorrect predictions (R/mR) on AeroEye is higher than PVSG (Tables 4 and 6), despite having more predicates (Table 1) and tiny objects. This suggests that our dataset is *less visually ambiguous* than PVSG.

# 2 Related Work

In this section, we review the existing datasets and benchmarks for Visual Scene Graph Generation, followed by a summary of the key challenges and issues related to Video Scene Graph Generation.

## 2.1 Visual Scene Graph Generation Datasets and Benchmarks

**Datasets.** VisDrone [17], DOTA [16], and SODA-A [36] image datasets, along with UAVid [46], UAVDT [37], and MAVREC [18] video datasets, offer high-resolution UAV datasets that enable precise object detection in dynamic scenes. While these UAV datasets focus on object detection, the Visual Genome [23] pioneered image-based SGG, and the Action Genome [6] dataset extended this concept to capture dynamic interactions within videos. Recently, ASPIRe [8] and SportsHHI [29] emphasize diverse human-object relationships and sports-specific player interactions. Additionally, PSG-4D [47] expands the VidSGG to encompass the 4D domain, bridging the gap between raw visual data and high-level understanding. In Table 1, we present a comparative overview of UAV-based and SGG datasets for images and videos, emphasizing their unique characteristics and advantages.

**Benchmarks.** The existing benchmark focuses on Image Scene Graph Generation (ImgSGG) and Video Scene Graph Generation (VidSGG). *ImgSGG* identifies and categorizes relationships between objects within an image into predefined relational categories, including Transformer-based

Table 1: Comparison of available datasets for scene graph generation. The top two blocks present image and video scene graph datasets, while the next two focus on image and drone video datasets. Viewpoints include five types: from ego (1st-person) view to 3rd-person view, and drone-captured perspectives, which is our main focus in this work (✓/✗ in colors), including aerial (top-down), oblique (slanted), and ground (eye-level) perspectives. # denotes the number of corresponding items. The best values in *drone* blocks are highlighted .

| Datasets | #Videos | #Frames | Resolution | Annotations | | | #ObjCls | #RelCls | #Scenes | Viewpoints | | | | |
|---|---|---|---|---|---|---|---|---|---|---|---|---|---|---|
| | | | | BBox | Relation | #Instances | | | | 3rd | ego | aerial | oblique | ground |
| Visual Genome [23] | - | **108K** | - | ✓ | ✓ | 3.8M | 33K | 42K | - | ✓ | ✗ | ✗ | ✗ | ✗ |
| VG-150 [24] | - | 88K | - | ✓ | ✓ | 2.8M | 150 | 50 | - | ✓ | ✗ | ✗ | ✗ | ✗ |
| VrR-VG [25] | - | 59K | - | ✓ | ✓ | 282.4K | 1.6K | 117 | - | ✓ | ✗ | ✗ | ✗ | ✗ |
| GQA [26] | - | 85K | - | ✓ | ✓ | | 1.7K | 310 | - | ✓ | ✗ | ✗ | ✗ | ✗ |
| PSG [27] | - | 49K | - | ✓ | ✓ | 538.2K | 80 | 56 | - | ✓ | ✗ | ✗ | ✗ | ✗ |
| VidVRD[28] | 1K | **296.2K** | 1920 × 1080 | ✓ | ✓ | 15.1K | 35 | 132 | - | ✓ | ✗ | ✗ | ✗ | ✗ |
| Action Genome [6] | 10K | 234.3K | 1280 × 720 | ✓ | ✓ | **476.3K** | 25 | 25 | - | ✓ | ✗ | ✗ | ✗ | ✗ |
| ASPIRe [8] | 1.5K | 1.6M | 1280 × 720 | ✓ | ✓ | 167.8K | 833 | 4.5K | 7 | ✓ | ✗ | ✗ | ✗ | ✗ |
| SportsHHI [29] | 80 | 11.4K | 1280 × 720 | ✓ | ✓ | 118.1K | 1 | 34 | 2 | ✓ | ✗ | ✗ | ✗ | ✗ |
| VidOR [30] | 10K | 55.4K | 640 × 360 | ✓ | ✓ | 50K | 80 | 50 | 1 | ✓ | ✗ | ✗ | ✗ | ✗ |
| VidSTG [31] | 10K | 55.4K | 640 × 360 | ✓ | ✓ | 50K | 80 | 50 | 1 | ✓ | ✗ | ✗ | ✗ | ✗ |
| EPIC-KITCHENS [32] | 700 | 11.5K | 1920 × 1080 | ✓ | ✓ | 454.3K | 21 | 13 | 1 | ✓ | ✓ | ✗ | ✗ | ✗ |
| PVSG [7] | 400 | 153K | 1920 × 1080 | ✓ | ✓ | 7.6K | 126 | 57 | 7 | ✓ | ✓ | ✗ | ✗ | ✗ |
| DOTA [16] | - | 11.3K | 1490 × 957 | ✓ | ✗ | 1.8M | 18 | - | - | ✗ | ✗ | ✓ | ✗ | ✗ |
| AI-TOD [33] | - | 28.1K | 800 × 800 | ✓ | ✗ | 700.6K | 8 | - | - | ✗ | ✗ | ✓ | ✗ | ✗ |
| DIOR-R [34] | - | 23.5K | 800 × 800 | ✓ | ✗ | 192.5K | 20 | - | - | ✗ | ✗ | ✓ | ✗ | ✗ |
| MONET [35] | - | **53K** | - | ✓ | ✗ | 162K | 3 | - | - | ✗ | ✗ | ✓ | ✗ | ✗ |
| SODA-A [36] | - | 2.5K | 4761 × 2777 | ✓ | ✗ | **872.1K** | 9 | - | - | ✗ | ✗ | ✓ | ✗ | ✓ |
| VisDrone [17] | 288 | 261.9K | 3840 × 2160 | ✓ | ✗ | **2.6M** | **10** | - | - | ✗ | ✗ | ✓ | ✗ | ✗ |
| UAVDT [37] | 100 | 40.7K | 1080 × 540 | ✓ | ✗ | 0.84M | 3 | - | 6 | ✗ | ✗ | ✓ | ✗ | ✗ |
| Stanford Drone [38] | **10K** | **929.5K** | - | ✓ | ✗ | - | 6 | - | 8 | ✗ | ✗ | ✓ | ✗ | ✗ |
| UIT-ADrone [39] | 51 | 206.2K | 1920 × 1080 | ✓ | ✗ | 210.5K | 8 | - | 1 | ✗ | ✗ | ✓ | ✗ | ✗ |
| ERA [40] | 2.9K | 343.7K | 640 × 640 | ✓ | ✗ | - | - | - | 25 | ✗ | ✗ | ✓ | ✗ | ✗ |
| MOR-UAV [41] | 30 | 10.9K | 1920 × 1080 | ✓ | ✗ | 89.8K | 2 | - | - | ✗ | ✗ | ✓ | ✗ | ✗ |
| AU-AIR [42] | - | 32.8K | 1920 × 1080 | ✓ | ✗ | 132K | 8 | - | - | ✗ | ✗ | ✓ | ✗ | ✗ |
| DroneSURF [43] | 200 | 411.5K | 1280 × 720 | ✓ | ✗ | 786K | 1 | - | - | ✗ | ✗ | ✗ | ✓ | ✗ |
| MiniDrone [44] | 38 | 23.3K | 224 × 224 | ✓ | ✗ | - | - | - | 1 | ✗ | ✗ | ✗ | ✓ | ✗ |
| Brutal Running [45] | - | 1K | 227 × 227 | ✓ | ✗ | - | - | - | 1 | ✗ | ✗ | ✗ | ✓ | ✗ |
| UAVid [46] | 30 | 300 | 4096 × 2160 | ✓ | ✗ | - | 8 | - | - | ✗ | ✗ | ✗ | ✓ | ✗ |
| MAVREC [18] | 24 | 537K | 3840 × 2160 | ✓ | ✗ | 1.1M | **10** | - | 4 | ✗ | ✗ | ✗ | ✓ | ✓ |
| AeroEye (Ours) | 2.3K | 261.5K | 3840 × 2160 | ✓ | ✓ | 1.8M | 57 | 384 | 29 | ✗ | ✗ | ✓ | ✓ | ✓ |

methods [27, 48, 49, 50] and generative-based models [51, 52, 53]. *VidSGG* leverages the dynamic nature of object interactions over time to better identify relationships, as the temporal dimension of videos provides a richer context for understanding semantic connections within the scene. Current methods using hierarchical structures [54, 8] or Transformer architectures [22, 55, 56, 57] excel at capturing long-range dependencies and complex interactions, advancing video understanding in video captioning[11, 12, 13], visual question answering [14, 15], and video grounding [58, 59, 60].

## 2.2 Video Scene Graph Generation

VidSGG can be categorized into two main types based on the granularity of its graph representation. ***Video-level SGG*** represents object trajectories as graph nodes, capturing constant relations between objects for a video. Various methods have been proposed to address this problem, incorporating Conditional Random Fields [61], abstracting videos [62], and iterative relation inference techniques [63] on fully connected spatio-temporal. However, focusing primarily on recognizing video-level relations directly based on object-tracking [64, 65, 66] results and neglecting frame-level scene graphs results in a cumbersome pipeline highly dependent on tracking accuracy. In contrast, ***Frame-level SGG*** defines the graph at the frame level, allowing relations to change over time. The releases of the benchmark datasets [67, 7, 8] have prompted the development of VidSGG models. TRACE [54], for instance, employs a hierarchical relation tree to capture spatio-temporal context information, while CSTTran [22] uses a spatio-temporal transformer to solve the problem. Recently, hierarchical interlacement graph (HIG) [8] abstracts relationship evolution using a sequence of hierarchical graphs.

## 2.3 Discussions

In this subsection, we conceptually compare our proposed approach with relationship modeling concepts discussed in Section 2.2 as illustrated in Fig. 2. In addition, we highlight the advantages of our approach and discuss the properties that distinguish it from these existing methods.

**Concepts.** The *progressive* approach fuses pairwise features between object pairs at each frame, encoding the object relationship at that specific step, followed by a fully connected layer to classify the predicate types. However, it processes frames independently without considering the temporal context. The *batch-progressive* approach employs a transformer block with positional embeddings on the fused query features. The *hierarchical* approach represents a video as a sequence of graphs, integrating temporal and spatial information at different levels. The node and edge features are updated at each hierarchical level based on the previous level to capture evolving object relationships.

**Limitations.** While the *batch-progressive* approach considers temporal information, both these *progressive* and *batch-progressive* approaches have limitations in modeling the full complexity of temporal dynamics and dependencies in the video. In contrast, the *hierarchical* graph approach can capture complex interactions and relationships between objects by considering the temporal evolution of graphs at different granularity levels. However, the hierarchical graph requires analyzing the entire video before constructing the graph (*i.e.* offline method). These limitations underscore the need for more advanced approaches to efficiently model temporal dynamics, adaptively update memory to handle evolving video data, and accurately capture the intricate relationships between objects.

**Advantages of Our Design.** Inspired by previous work [68, 69], which processes temporal features through iterative feedback loops and circular updating, we propose the CYCLO approach that circularly incorporates an updated history of relationships. In contrast to these methods, which focus on frame-level updates influenced by global features, our approach constructs and refines scene graphs for each frame, capturing static spatial relationships between objects and their dynamic evolution over time. By leveraging circular connectivity, CYCLO establishes a continuous loop of temporal information, ensuring no temporal edge is treated as a boundary. It enables the Transformer to operate online and then capture and update relationships between objects more effectively, correcting erroneous connections. The theoretical properties are included in Section B of Appendices.

## 3  The Proposed AeroEye Dataset

In this section, we detail the AeroEye dataset annotation process and provide the dataset statistics.

### 3.1  Dataset Collection

**Data Preparation.** We leverage videos from the ERA [40] and MAVREC [18] datasets to construct our AeroEye dataset. ERA consists of diverse scenes ranging from rural to urban environments in extreme conditions (*e.g.* earthquake, flood, fire, mudslide), daily activities (*e.g.* harvesting, plowing, party, traffic collision), and sports activities (*e.g.* soccer, basketball, baseball, running, swimming). MAVREC features sparse and dense object distributions and contains typical outdoor activities characterized by many vehicle classes, incorporating viewpoint changes and varying illumination.

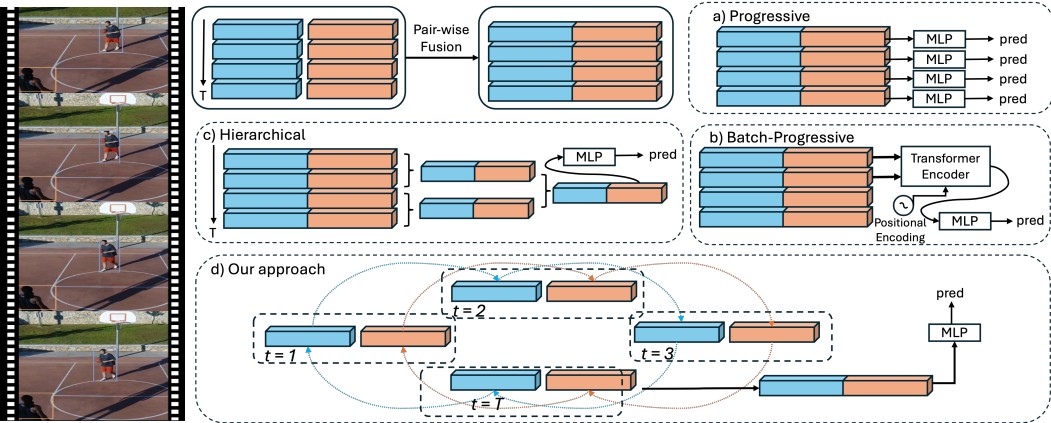

Figure 2: Comparisons of CYCLO and existing relationship modeling: (a) *Progression* [27, 63]: frame-wise fusion and classification; (b) *Batch-progression* [22, 55, 56]: temporal transformer; (c) *Hierarchy* [8]: spatiotemporal graph; (d) Our CYCLO approach: circular connectivity for capturing temporal dependencies.

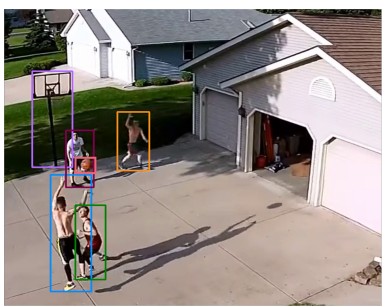
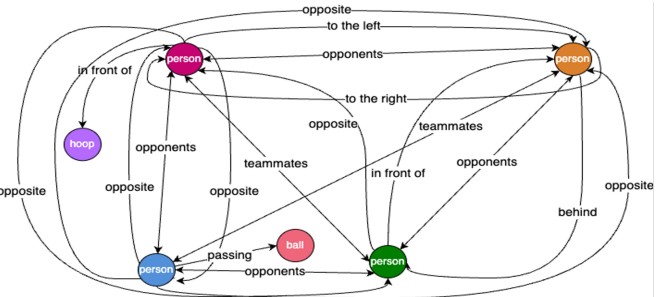

a) Basketball scene from the ERA dataset    b) A scene graph depicting our annotations of relationships between objects.

Figure 3: Example annotation in our dataset. In Fig. 3b, straight arrows denote relationships between objects, while curved arrows indicate the positions of the objects. Nodes of the same color represent the same object, and the labels on the edges specify the predicate of each relationship. (Best viewed in colors)

**Relationship Classes and Instance Formulation.** In Fig. 3, we focus on two aspects of object relationships: positions (*e.g.* in front of, behind, next to) and relations, which consist of movement actions (*e.g.* chasing, towing, overtaking) and collision actions (*e.g.* hitting, crashing, colliding). These relationships are semantically complex and require detailed spatio-temporal context reasoning for recognition. Following previous PVSG benchmark datasets [6, 7, 8], we define relationship instances at the frame level, considering the long-term spatial-temporal context. Each instance is formulated as a triplet $<s, o, p>$, where $s$ and $o$ denote the bounding boxes of the subject and object, and $p$ represents the predicate (*i.e.* position and relation), included in Tables A.8, A.9 which are summarized in Fig. 4. In addition, Fig. A.11 presents selected samples from our AeroEye dataset.

## 3.2 Data Specification

**Data Annotation.** We annotate keyframes at 5FPS to capture frequent and rapid changes in positions and relations in aerial videos, reducing redundancy while keeping up with interaction changes. Our two-stage annotation pipeline first performs *object localization and tracking* and then *relationship instance annotation*. To generate diverse predicates, we leverage the GPT4RoI [70] model, which combines visual and linguistic data to generate detailed

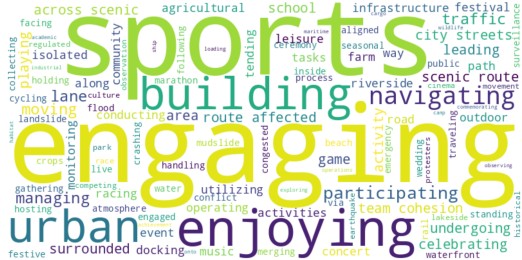

Figure 4: Relationship word cloud on AeroEye dataset.

descriptions of object relationships within specified regions of interest. Although we annotate relationship instances frame by frame, as illustrated in Fig. 3, we easily create relationship tubes using the provided object tracking ID by connecting the same pair of objects with the same relationship predicate across consecutive frames. The annotation file includes object information (*i.e.*, bounding boxes, category names, and tracking IDs) and relationships within each frame. Details on the quality control process and annotation examples can be found in Section A.2 of the Appendices.

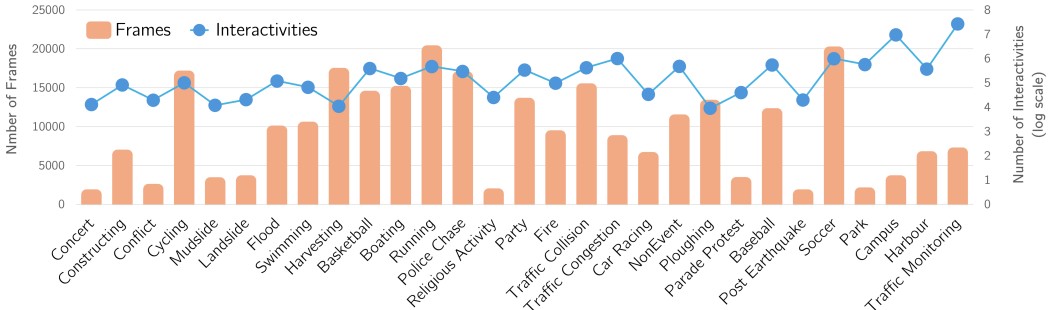

Figure 5: Statistics for each scene on the AeroEye dataset.

**Data Statistics.** The AeroEye dataset is a collection of 2,260 videos with 261,503 frames, annotated with over 2.2 million bounding boxes across 56 object categories typically observed from *aerial*, *oblique*, and *ground* perspectives captured by drone. Specifically, our AeroEye dataset consists of 384 predicates divided into two relationship categories: 135 positions and 249 relations. The key strength of AeroEye is its richness in relationships. On average, each video in the dataset has 127 frames, providing moderate temporal depth for capturing detailed interactions. The average number of frames per scene is 8,970, indicating substantial variability and complexity. AeroEye is rich in relationships, with more than 43 million relationship instances. In Table 1, we provide a detailed comparison with related datasets, while Fig. 5 presents statistics on the AeroEye dataset. In addition, predicate definitions and additional statistics are discussed in Sections A.1 and A.4 of the Appendices.

## 4 The Proposed Approach

In this section, we present our CYCLO approach, including the *Spatial Attention Graph* and the *Cyclic Temporal Graph Transformer*. The *Spatial Attention Graph* captures spatial dependencies and interactions between objects within the frame. In contrast, the *Cyclic Temporal Graph Transformer* models temporal relationships across frames, capturing short-term and long-term dynamics.

### 4.1 Problem Formulation

Given an input video with $T$ frames, we construct dynamic scene graphs $\{\mathcal{G}_t\}_{t=1}^{T}$ that encode the relationships among objects within these frames. Each graph $\mathcal{G}_t(\mathcal{V}_t, \mathcal{E}_t)$ captures static relationships, where node $\mathcal{V}$ consists of objects and edge $\mathcal{E}$ denotes the relationship between objects. Each object $v_i \in \mathcal{V}$ has an object category $v_i^c \in \mathcal{C}_v$ and box coordinates $v_i^b \in [0,1]^4$. Each relationship $e_j \in \mathcal{E}$ represents the $j$-th triplet $(s_j, o_j, p_j)$, where subject $s_j$ and object $o_j$ and predicate $p_j \in \mathcal{C}_p$.

### 4.2 Spatial Attention Graph

Self-attention mechanisms in one-stage object detectors [71, 19] model relationships between objects, allowing insights into the dynamics between entities without relying on additional contextual information. For example, in an aerial parking lot video with cars, vans, and people, if the self-attention layer strongly connects the queries representing the person and the car, it suggests an interaction, such as the person entering the vehicle. Inspired by [50], in our CYCLO approach, to construct the static graph in each frame $t$, we utilize the DETR decoder to establish bidirectional connections among object queries. In particular, we compute relational representations at each layer $l$ by concatenating the query and key vectors, $Q_t^l$ and $K_t^l$, for every object pair. This process ensures the layer $l-1$ output seamlessly transitions as input to layer $l$. We omit the superscript $L$ related to the final layer to simplify the following discussion. At each frame, $\widehat{R}_{a,t}^l$ captures the dynamic interplay of relations at layer $l$, utilizing their query and key vectors. Furthermore, $\widehat{R}_{z,t}$ leverages object queries in the final layer for object detection. These relationships are formally defined in Eqn. (1).

$$\widehat{R}_{a,t}^l = [Q_t^l \phi_{W_s^l}; K_t^l \phi_{W_o^l}], \quad \widehat{R}_{z,t} = [\widehat{Z}_t \phi_{W_s}; \widehat{Z}_t \phi_{W_o}] \tag{1}$$

Here, $\phi_{W_s}$ and $\phi_{W_o}$ are the linear transformations that process subject and object features, enabling the model to consider both object characteristics and their interrelationships comprehensively. In addition, gating mechanisms $g_{a,t}^l$ and $g_{z,t}$ dynamically modulate the contributions from different layers. These gated representations from all layers are then integrated to construct a relation matrix:

$$\widehat{g}_{a,t}^l = \sigma(\widehat{R}_{a,t}^l \phi_{W_G}), \quad \widehat{g}_{z,t} = \sigma(\widehat{R}_{z,t} \phi_{W_G})$$
$$\widehat{R}_t = \sum_{l=1}^{L} (\widehat{g}_{a,t}^l \times \widehat{R}_{a,t}^l) + \widehat{g}_{z,t} \times \widehat{R}_{z,t} \tag{2}$$

where $\phi_{W_G}$ denotes the linear weight applied during the gating process. Finally, the relation matrix is fed into the three-layer perception (MLP) with ReLU activation and a sigmoid function $\sigma$, which predicts multiple relationships $(p_j)$ between pairs of objects $(s_j, o_j)$. Mathematically, $\widehat{G}_t = \sigma(\text{MLP}(\widehat{R}_t))$ is the graph representation at frame $t$-th, where $\widehat{G}_t \in \mathbb{R}^{N \times N \times |\mathcal{C}_p|}$.

**Discussion.** Transformer-based approaches to VidSGG effectively capture interactions and temporal changes through self-attention mechanisms, creating detailed scene graphs that reflect video dynamic

relationships. However, these models often struggle to represent the directional and historical aspects of the relationship accurately. While effective at identifying token correlations, the scaled dot-product fails to consider their temporal or spatial ordering. This oversight is particularly critical in videos, where understanding the historical context of relationships is essential. For example, the sequence of relationships leading up to a car crash, including speeding, lane changing, and passing, must be considered. Each interaction change provides crucial historical information that contextualizes the final relationships, vital for enhancing prediction and interpretation. In addition, traditional selft-attention [72] does not adequately capture this essential sequential, *directional*, and *historical information* [73]. It highlights the need for advancements in transformer architectures to more effectively integrate the direction and historical sequence of interactions within videos.

## 4.3   Cyclic Temporal Graph Transformer

We present the Cyclic Spatial-Temporal Graph Transformer to refine the spatial attention graph in each scene, capturing temporal dependencies via subject-object relationships across adjacent frames.

**Cyclic Attention.** As mentioned in Section 4.2, self-attention does not adequately capture *directional* and *historical information*. Therefore, we propose the cyclic attention (CA), defined as in Eqn. (3).

$$\text{CA}(Q_t, K_t) = \sum_{i=0}^{T-1} \sigma \left( \frac{Q_t(K_{\eta(t+i) \bmod T})^\top}{\sqrt{d_{head}}} \right) \tag{3}$$

In Eqn. (3), $\eta$ is a shift term enabling cyclical indexing via $\bmod T$. The cyclical indexing, illustrated in Fig. 6, allows for continuous sequence processing by connecting the end to the beginning, which is crucial for predicting movements in dynamic interactions where past events influence future actions (*e.g.* a car navigating a roundabout). Cyclical indexing differs from standard self-attention as it is a permutation equivariant without positional encodings. In contrast, cyclical attention is non-permutation equivariant, which depends on the original sequence order. This property is crucial for multi-object relationship modeling, where maintaining the chronological order of interactions is essen-

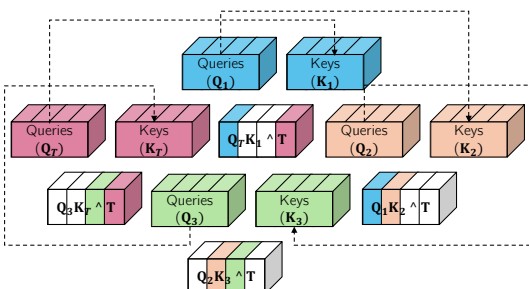

Figure 6: Illustration of cyclic interactions in the Cyclic Spatial-Temporal Graph Transformer. Each frame, represented by a colored block (where the first frame, $t = 1$ and the last frame, $T = 4$), undergoes spatial attention to obtain queries ($Q_t$) and keys ($K_t$).

tial. For instance, in a surveillance scenario, the sequence of a car stopping for a pedestrian must preserve the order of events, ensuring the vehicle stops before the pedestrian appears.

**Temporal Graph Transformer.** Our Temporal Graph Transformer refines spatial attention graphs, $\{\widehat{G}_t\}_{t=1}^T$, into a sequence of dynamic graphs $\{\mathcal{G}_t\}_{t=1}^T$, leveraging the temporal dynamics and spatial interactions of objects across video frames. Our approach employs a series of cyclic attention blocks configured within multi-head attention to refine object representations by integrating features from adjacent frames. The core of our approach is the integration of cyclic attention into a multi-head structure, which processes the sequence of input features $\widehat{Z} = \{\widehat{Z}_t\}_{t=1}^T$, represented as in Eqn. (4).

$$Z' = \phi_{W_c}([h_0; h_1; \dots; h_{e-1}]), \quad Z'_t = \phi_{W_c}([h_0(t); h_1(t); \dots; h_{e-1}(t)]),$$
$$h_i(t) = \text{CA}(\phi_{W_q^i}(\widehat{Z}_t), \phi_{W_k^i}(\widehat{Z})), \quad i \in \{0, 1, \dots, e-1\}, \tag{4}$$

where $\phi_{W_c}$, $\phi_{W_q^i}$, and $\phi_{W_k^i}$ denote the linear transformations. Each head $h_i(t)$ computes the cyclic attention, integrating information across the video to enhance the temporal relationship at each frame. The outputs from various heads at each frame are integrated into $Z'_t$, derived from concatenating all attention head outputs. These heads process features cyclically across different representation subspaces to capture the temporal evolution of relationships in the video. Then, $Z$ is obtained by applying layer normalization (LN) and a skip connection to the aggregated features $Z'_t$, where $Z = \text{LN}(Z' + \widehat{Z})$. This step ensures that $Z'_t$ is stabilized and effectively integrated with the original features $\widehat{Z}$, thus dynamically updating the scene graph and ensuring temporal coherence.

In addition, $Z_t$ is utilized to construct a new relation matrix $R_t$ by applying the transformations in Eqn. (1) and (2). This updated matrix $R_t$ refines the relationship dynamics captured in static frames

Table 2: Our performance (%) on AeroEye with shift values ($\eta$ in Eqn. (3)) at Recall (R) and mean Recall (mR).

| Shift Value | PredCls | | | SGCls | | | SGDet | | |
|---|---|---|---|---|---|---|---|---|---|
| | R/mR@20 | R/mR@50 | R/mR@100 | R/mR@20 | R/mR@50 | R/mR@100 | R/mR@20 | R/mR@50 | R/mR@100 |
| 1 | **56.20 / 19.23** | **61.62 / 20.67** | **62.40 / 21.19** | **54.15 / 16.22** | **59.59 / 18.20** | **60.37 / 18.38** | **43.53 / 13.29** | **47.93 / 13.69** | **48.94 / 13.86** |
| 2 | 55.01 / 18.01 | 60.02 / 19.02 | 61.03 / 19.53 | 53.04 / 15.05 | 58.06 / 17.07 | 59.08 / 17.25 | 42.09 / 12.10 | 46.11 / 12.56 | 47.12 / 12.78 |
| 3 | 54.12 / 17.11 | 59.13 / 18.12 | 60.14 / 18.63 | 52.17 / 14.18 | 57.19 / 16.20 | 58.21 / 16.42 | 41.28 / 11.30 | 45.30 / 11.82 | 46.32 / 12.04 |
| 4 | 54.55 / 17.55 | 59.56 / 18.56 | 60.57 / 19.07 | 52.59 / 14.60 | 57.61 / 16.62 | 58.63 / 16.84 | 41.75 / 11.76 | 45.77 / 12.28 | 46.79 / 12.50 |
| 5 | 54.33 / 17.34 | 59.35 / 18.36 | 60.37 / 18.88 | 52.40 / 14.41 | 57.42 / 16.43 | 58.44 / 16.65 | 41.50 / 11.51 | 45.53 / 12.03 | 46.55 / 12.26 |

Table 3: Our performance (%) on AeroEye for varying frames per video at Recall (R) and mean Recall (mR).

| # Frame Discarded | PredCls | | | SGCls | | | SGDet | | |
|---|---|---|---|---|---|---|---|---|---|
| | R/mR@20 | R/mR@50 | R/mR@100 | R/mR@20 | R/mR@50 | R/mR@100 | R/mR@20 | R/mR@50 | R/mR@100 |
| 1 | **56.20 / 19.23** | **61.62 / 20.67** | **62.40 / 21.19** | **54.15 / 16.22** | **59.59 / 18.20** | **60.37 / 18.38** | **43.53 / 13.29** | **47.93 / 13.69** | **48.94 / 13.86** |
| 2 | 55.08 / 18.82 | 60.39 / 20.26 | 61.15 / 20.76 | 53.07 / 15.90 | 58.40 / 17.84 | 59.16 / 18.01 | 42.66 / 13.02 | 46.97 / 13.41 | 47.96 / 13.58 |
| 3 | 53.98 / 18.42 | 59.18 / 19.85 | 59.93 / 20.34 | 51.99 / 15.58 | 57.23 / 17.49 | 57.97 / 17.65 | 41.81 / 12.75 | 46.03 / 13.14 | 47.00 / 13.30 |
| 4 | 52.90 / 18.04 | 57.99 / 19.46 | 58.73 / 19.93 | 50.95 / 15.27 | 56.08 / 17.14 | 56.81 / 17.30 | 40.97 / 12.49 | 45.11 / 12.87 | 46.06 / 13.03 |
| 5 | 51.84 / 17.66 | 56.83 / 19.08 | 57.55 / 19.53 | 49.93 / 14.96 | 54.96 / 16.80 | 55.68 / 16.96 | 40.17 / 12.24 | 44.23 / 12.61 | 45.14 / 12.76 |

by correcting spurious or incomplete relationships and incorporating previously omitted ones using the temporal context from the frame sequence. As a result, $G_t$ comprehensively represents persistent and transient interactions, including their direction and historical sequence within the video.

**Loss Function.** Visual object relationships involve predicates that may appear quite similar, such as "parking next to" and "stopping next to". Thereby, we utilize a multi-label margin loss, as in [22]:

$$\mathcal{L}_p(r, \mathcal{P}^+, \mathcal{P}^-) = \sum_{p \in \mathcal{P}^+} \sum_{q \in \mathcal{P}^-} \max(0, 1 - \phi(r, p) + \phi(r, q)) \tag{5}$$

In Eqn. (5), $r$ represents a subject-object pair, and $\mathcal{P}^+$ and $\mathcal{P}^-$ correspond to positive and negative predicates, respectively. The term $\phi(r, p)$ measures the compatibility of the pair $r$ with the predicate $p$. Additionally, object distributions are modeled using neural networks with ReLU activation and batch normalization. Cross-entropy loss is applied during the learning process. The total loss is a combination of the multi-label margin loss $\mathcal{L}_p$ and the cross-entropy loss $\mathcal{L}_{ce}$, defined as:

$$\mathcal{L}_{total} = \mathcal{L}_p + \lambda \mathcal{L}_{ce} \tag{6}$$

where $\lambda$ is a weight balancing the contribution of the cross-entropy loss $\mathcal{L}_{ce}$.

# 5 Experimental Results

In this section, we discuss the benchmark dataset evaluations and comparisons with SOTA methods.

## 5.1 Implementation Details

**Dataset.** We use 10-fold cross-validation on the AeroEye dataset, including 1,797 videos for training and 463 videos for testing. We also evaluate our performance on PVSG [7] and ASPIRe [8] datasets.

**Settings.** We employ DINO [19] to extract the spatial attention graphs (in Section 4.2). DINO is trained with ResNet-50 backbone and 1500 queries on MAVREC, achieving 92.35 mAP on the validation set. The pre-trained detector is applied to baselines, and parameters are fixed during subsequent task training. Our model is trained on $8 \times$ A6000 GPUs using 12 epochs with AdamW optimizer (initial learning rate of $1e^{-5}$ and a batch size of 1), gradient clipping (max norm of 5).

**Evaluation Metrics.** We evaluate models on two standard tasks in image-based scene graph generation followed by previous work [74, 7] that are predicate classification (*PredCls*), scene graph classification (*SGCls*), and scene graph detection (*SGDet*). While *SGCls* predicts relationships given ground truth objects, *SGDet* involves detecting objects and predicting relationships. These tasks are evaluated using Recall (R@K) and mean Recall (mR@K), where $K \in \{20, 50, 100\}$.

## 5.2 Ablation Study

**Semantic Dynamics in Cyclic Attention.** By altering $\eta$ (in Eqn. (3)), we consider the permutation or non-equivariance equivariance. If the predictions systematically adapt to the shifts induced by

Table 4: Comparison (mean ± std) on AeroEye against baseline methods in terms of *Recall* (R).

| Method | PredCls | | | SGCls | | | SGDet | | |
|---|---|---|---|---|---|---|---|---|---|
| | R@20 | R@50 | R@100 | R@20 | R@50 | R@100 | R@20 | R@50 | R@100 |
| **Vanila (2a)** | 50.12 ± 1.80 | 54.68 ± 5.70 | 56.32 ± 3.45 | 48.09 ± 1.78 | 53.23 ± 5.66 | 54.99 ± 3.38 | 31.04 ± 3.41 | 34.28 ± 0.72 | 34.62 ± 1.10 |
| **Transformer (2b)** | 53.25 ± 1.71 | 59.35 ± 4.02 | 60.89 ± 0.88 | 51.12 ± 1.66 | 57.12 ± 3.91 | 59.19 ± 0.83 | 41.09 ± 0.42 | 46.52 ± 0.63 | 47.15 ± 0.83 |
| **HIG (2c)** | 54.18 ± 1.23 | 59.59 ± 5.90 | 60.35 ± 5.30 | 52.03 ± 1.19 | 57.47 ± 5.87 | 58.18 ± 5.22 | 37.28 ± 0.60 | 38.59 ± 2.37 | 39.27 ± 1.49 |
| **CYCLO (2d - Ours)** | **56.20 ± 0.70** | **61.62 ± 2.90** | **62.40 ± 1.88** | **54.15 ± 0.67** | **59.59 ± 2.82** | **60.37 ± 1.83** | **43.53 ± 0.30** | **47.93 ± 0.65** | **48.94 ± 0.79** |

Table 5: Comparison (mean ± std) on AeroEye against baseline methods in terms of *mean Recall* (mR).

| Method | PredCls | | | SGCls | | | SGDet | | |
|---|---|---|---|---|---|---|---|---|---|
| | mR@20 | mR@50 | mR@100 | mR@20 | mR@50 | mR@100 | mR@20 | mR@50 | mR@100 |
| **Vanila (2a)** | 12.21 ± 0.51 | 13.34 ± 0.62 | 13.58 ± 0.73 | 8.05 ± 0.42 | 8.15 ± 2.65 | 8.77 ± 1.63 | 11.20 ± 2.38 | 11.27 ± 0.84 | 12.43 ± 0.68 |
| **Transformer (2b)** | 14.25 ± 0.46 | 15.78 ± 0.53 | 16.24 ± 0.58 | 11.12 ± 0.35 | 13.12 ± 1.53 | 13.69 ± 1.32 | 11.88 ± 0.05 | 12.31 ± 0.13 | 12.91 ± 0.15 |
| **HIG (2c)** | 18.37 ± 0.68 | 19.85 ± 0.79 | 20.43 ± 0.88 | 15.10 ± 0.37 | 17.08 ± 2.66 | 17.69 ± 1.79 | 11.97 ± 1.07 | 13.12 ± 2.43 | 13.29 ± 2.41 |
| **CYCLO (2d - Ours)** | **19.23 ± 0.32** | **20.67 ± 0.42** | **21.19 ± 0.48** | **16.22 ± 0.04** | **18.20 ± 1.29** | **18.38 ± 0.43** | **13.29 ± 0.46** | **13.69 ± 0.53** | **13.86 ± 0.55** |

different $\eta$ values, it demonstrates a degree of *permutation equivariance*. Conversely, if the predictions change in ways that do not correspond systematically to these shifts, it may indicate *non-permutation equivariance*. Table 2 shows a decrease in performance, implying disrupted temporal patterns and *non-permutation invariance*, indicated by unpredictable output changes relative to input shifts.

**Cyclic Dependency.** To further validate the *cyclic* dependency of our model, we discarded frames from every successive frame. Removing frames disrupts the temporal continuity of the sequence, which is crucial for maintaining the the cyclic nature of video. If the cyclic model presupposes that each frame has direct relationships with its adjacent frames circularly, then removing frames could sever these relationships, potentially diminishing the ability to leverage cyclical patterns effectively. Indeed, Table 3 informs a decrease in Recall and mean Recall when frames were reduced.

## 5.3 Comparisons with Baseline Methods

Table 4 shows CYCLO outperforms other methods across metrics and tasks, surpassing Transformer by 2.95%, 3.03%, and 2.44% in the PredCls, SGCls, and SGDet at R@20, and maintaining its lead even at higher Recall thresholds. Moreover, Table 5 shows significant improvements at mR@20, mR@50, and mR@100, with CYCLO leading HIG by 0.86% in the PredCls task at mR@20 and exceeding HIG by 1.12% and 1.32% in the SGCls and SGDet tasks, respectively, at mR@20. In addition, the consistent performance and low standard deviation of the CYCLO model across Table 4 and 5 demonstrate its robustness and overall superiority. Fig. 7 displays that CYCLO can capture the evolving relationships between objects by updating their positions and interactions.

## 5.4 Comparisons with State-of-the-Art Methods

We compare our performance on two recent benchmark datasets on VidSGG (*i.e.* PVSG and ASPIRe).

Table 6: Comparative performance (%) of our model and previous methods on the PVSG dataset, evaluated by Recall (R) and mean Recall (mR).

| Model | R/mR@20 | R/mR@50 | R/mR@ 100 |
|---|---|---|---|
| **Vanilla (2a)** | 2.35 / 1.22 | 2.71 / 1.31 | 2.94 / 1.45 |
| **Handcrafted (2b)** | 2.56 / 1.24 | 2.78 / 1.35 | 3.05 / 1.54 |
| **1D Convolution (2b)** | 2.79 / 1.24 | 2.80 / 1.47 | 3.10 / 1.59 |
| **Transformer (2b)** | 4.02 / 1.75 | 4.41 / 1.86 | 4.88 / 2.03 |
| **HIG (2c)** | 4.60 / 1.89 | 4.88 / 2.05 | 5.43 / 2.23 |
| **CYCLO (2d - Ours)** | **5.83 / 1.98** | **6.12 / 2.15** | **6.70 / 2.34** |

Table 7: Comparative performance (%) of our model and previous methods on the ASPIRe dataset, evaluated by Recall (R) and mean Recall (mR).

| Model | Interactivity | R/mR@20 | R/mR@50 | R/mR@ 100 |
|---|---|---|---|---|
| **Vanilla (2a)** | *Position* | 10.52 / 0.50 | 21.97 / 0.55 | 38.05 / 0.62 |
| | *Relation* | 9.71 / 0.32 | 21.96 / 0.36 | 39.11 / 0.40 |
| **Handcrafted (2b)** | *Position* | 10.73 / 0.52 | 22.04 / 0.59 | 38.16 / 0.71 |
| | *Relation* | 9.92 / 0.34 | 22.03 / 0.40 | 39.22 / 0.49 |
| **1D Convolution (2b)** | *Position* | 10.96 / 0.52 | 22.06 / 0.71 | 38.21 / 0.76 |
| | *Relation* | 10.15 / 0.34 | 22.05 / 0.52 | 39.27 / 0.54 |
| **Transformer (2b)** | *Position* | 11.04 / 0.83 | 22.52 / 0.90 | 38.84 / 1.02 |
| | *Relation* | 10.23 / 0.65 | 22.51 / 0.71 | 39.90 / 0.96 |
| **HIG (2c)** | *Position* | 13.02 / 0.09 | 24.52 / 1.33 | 42.33 / 1.12 |
| | *Relation* | 10.26 / 0.29 | 23.72 / 0.34 | 41.47 / 0.39 |
| **CYCLO (2d - Ours)** | *Position* | **13.71 / 0.85** | **26.07 / 1.45** | **43.94 / 1.49** |
| | *Relation* | **15.29 / 0.84** | **24.95 / 1.61** | **46.44 / 1.52** |

**Performance on *PVSG*.** In Table 6, we report that the CYCLO approach achieves superior performance compared to other models on the PVSG dataset, particularly in terms of Recall and mean Recall metrics. In particular, CYCLO surpasses HIG and Transformer by 1.23% and 1.81% at R@20, respectively. In addition, it consistently outperforms both models at higher recall rates, including R@50 and R@100, showcasing its effectiveness across various thresholds. Our approach also slightly

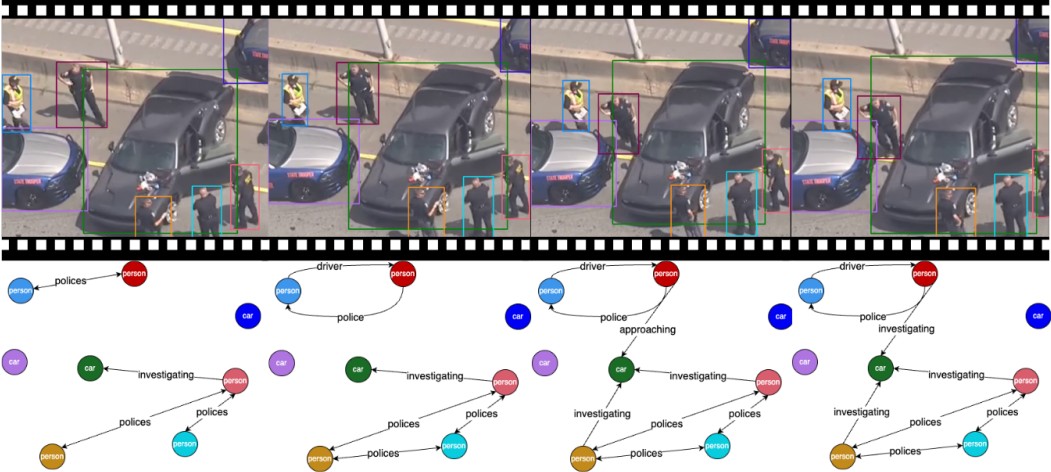

Figure 7: Scene graphs generated by the CYCLO model on the AeroEye dataset, illustrating dynamic relationships between objects and agents across UAV-captured frames. (Best viewed in colors)

improves the mean Recall, as the experimental results demonstrate. These results highlight the robustness of our CYCLO approach in recognizing and handling periodic actions, such as cooking, washing, cleaning, exercising, and other routine tasks frequently represented on the PVSG dataset.

**Performance on *ASPIRe*.** The ASPIRe dataset includes five distinct interactivity types. However, we focus on position and relation to ensure a fair comparison. As shown in Table 7, CYCLO consistently outperforms existing models across multiple recall and mean recall thresholds. Notably, at R@20, our CYCLO approach outperforms the HIG method, the second-best model, by 0.69% in position and a more substantial 5.03% in relation. Additionally, CYCLO shows significant gains in mean Recall across all evaluated thresholds, demonstrating its effectiveness in tackling the long-tail distribution.

## 6 Conclusions

We have introduced CYCLO, a novel approach that effectively captures periodic and overlapping relationships, handles extended sequences, and minimizes information loss, making it suitable for complex temporal modeling. In addition, we presented AeroEye, a comprehensive and diverse dataset composed of drone-captured scenes, specifically designed to represent intricate object relationships and spatial positions in aerial videos. Through extensive experiments on the AeroEye dataset and two large-scale in-the-wild datasets (*i.e.* ASPIRe and PVSG), we demonstrated the robustness and effectiveness of CYCLO in capturing dynamic interactions and evolving relationships over time.

**Limitations.** Although our CYCLO approach has achieved impressive performance, it may reveal limitations when dealing with incomplete or discontinuous videos. The periodic and cyclic attention mechanisms, crucial for capturing temporal and spatial object relationships, heavily rely on video continuity and completeness. Interruptions in the sequence, such as missing or discontinuous frames, disrupt the formation of accurate cyclical references, leading to inconsistent and incorrect predictions.

**Broader Impacts.** The proposed approach improves the capture of object interactions and temporal evolution in aerial and in-the-wild videos, which is critical for surveillance, disaster response, traffic management, and precision agriculture applications. By modeling object interactions over time, CYCLO supports more informed decision-making, leading to safer and more sustainable practices. This advancement opens the door for future work developing surveillance systems that can model complex relationships from drone videos. However, it is important to recognize the potential risks associated with this approach, particularly the possibility of using it for unauthorized surveillance.

**Acknowledgment.** This work is partly supported by J.B. Hunt Transport Services (JBHunt), NSF Data Science and Data Analytics that are Robust and Trusted (DART), NSF SBIR Phase 2, and Arkansas Biosciences Institute (ABI) grants. We also acknowledge Thanh-Dat Truong for invaluable discussions and the Arkansas High-Performance Computing Center (AHPCC) for providing GPUs.

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

# Appendices

## A  The AeroEye Dataset

### A.1  Relationship Definition

Our AeroEye dataset focuses on capturing the spatial positions and relationships between objects, including person-to-object and object-to-object interactions. Table A.8, A.9 provides a comprehensive list of the defined positions and relationships. Furthermore, Table A.10 offers guidance on the final relationship vocabulary specific to each scene category within the dataset.

Table A.8: A list of the 135 position predicates defined on the AeroEye dataset.

| | | |
|---|---|---|
| above | at focal points | near playground equipment |
| above debris | at intersection | near safe exits |
| above water lines | at the ends of the pool | near scoring zones |
| across different terrains | at the forefront of movements | near storage areas |
| across multiple lanes | at the helm | next to |
| across negotiation tables | at the junction | on benches |
| across ploughed fields | at the starting line | opposite |
| across the court | at the stern or sides | oriented |
| across the site | at turning point | outside |
| adjacent to | behind | outside danger zones |
| ahead in the race | behind agricultural machinery | outside of traffic flow |
| ahead of competitors | behind the congregation | over |
| aligned on the track | below unstable hillsides | over open waters |
| along a designated route | beneath | parallel |
| along cleared pathways | beside | parallel in lanes |
| along docks and aboard ships | beside bike lanes | surrounding the field |
| along monitored sections | beside debris | through city streets |
| along paths | beside merchandise stands | through debris |
| along pit stops | beside pool edges | through rows of crops |
| along roads | beside the congregation | throughout the fields |
| along the field | between | to the left |
| along the route | between buildings | to the right |
| along the sidelines | between conflicting parties | together |
| along the sides | between crop rows | under |
| along the sides of the pool | centered | undergoing |
| along viewing areas or walkways | close by | underneath |
| alongside construction machinery | close to | within affected zones |
| amidst debris | closely positioned | within audience clusters |
| angled | clustered | within bottleneck points |
| around campus landmarks | distant | within cabin areas |
| around common areas | down | within crash sites |
| around damaged structures | facing | within crowd formations |
| around familiar objects | facing the audience | within dance areas |
| around heavy equipment | facing the stage | within debris piles |
| around penalty boxes | from a vantage point | within designated lanes |
| around refreshment tables | from central points | within discussion circles |
| around ritual objects | from control centers | within goal areas |
| around stage areas | in designated pit areas | within marked lanes |
| around temporary aid stations | in front of | within moored boats |
| around the deck | in lane | within open fields |
| around the fire | indifferent | within outfield areas |
| around the vehicles | inside | within reach of fire hydrants |
| around water features | interwoven | within spectator areas |
| at a safe distance | near | within surveillance zones |
| at batting positions | near debris | within the basketball court |

Table A.9: A list of the 249 relationship predicates defined on the AeroEye dataset.

| | | | |
|---|---|---|---|
| acquaintances | dribbling | leading the way | riding |
| adhering to | drivers | learning | runners |
| alerting | encircled by | learning about farming | running |
| aligned in traffic lane | encircling | living in coastal community | scoring |
| aligned movement | enclosed | loading | selling |
| aligning in lane | engaging in beach sports | loading cargo onto ship | selling merchandise |
| along sidewalk | engaging in leisure activity | managing airport operations | setting up |
| along the lane | engaging in school sports | managing industrial operations | setting up temporary shelters |
| along the road | engaging in winter activities | managing urban traffic | sharing lane |
| along the sidewalk | enjoying concert atmosphere | merging lane | sharing slogans |
| amidsting traffic | enjoying lakeside park | mimicking | sharing their excitement |
| appreciating art exhibits | enjoying live music | monitoring | shopping |
| approaching | enjoying outdoor cinema | motivating | shopping in city |
| approaching intersection | enjoying outdoor festivities | moving | showcasing classic car |
| at center of intersection | enjoying spring festivities | moving at intersection | showcasing local art |
| at the conjunction | enjoying waterfront gathering | moving towards | singing |
| at the stop sign | enjoying wilderness | navigating | skating |
| attending games | enjoying winter recreation | negotiating | skiing |
| boarding | enjoying winter sports | observing | socializing |
| boaters | entertaining outdoor crowd | operating in docking area | standing inside |
| bordering | excavating | operating market stalls | standing outside |
| building coastal protection | exchanging | opponents | stopping at |
| building team cohesion | exercising | overlooking | straddling lanes |
| building underwater structure | exploring forest trails | overtaking | surrounded by |
| building urban infrastructure | exploring mountain trails | owner | swimmers |
| camping | exploring urban landscapes | painting | tackling |
| capturing | exploring wildlife habitat | parallel | teammates |
| capturing urban night lights | external to the building | parking at | tending to agricultural tasks |
| catching | extinguishing | participating in community event | touring |
| celebrating | farmer and harvesting tool | participating in festivities | toward destination |
| celebrating in festive atmosphere | farmers | participating in marathon | towing |
| celebrating public festival | finding parking lot | participating in music camp | tracking |
| changing direction | firefighters | partners | trading |
| chasing | floating | passing | training |
| cheering | following behind | performing | traveling |
| cleaning | forefronted | picnicking together | traversing |
| close friends | friends | players | trying to mediate resolutions |
| co-traveling | friends dancing together | playing | under observation |
| coaching | gathering | police | under surveillance |
| coincidental | graduating | practicing | undergoing |
| collaborating | handling emergency | preparing for emergencies | undergoing process |
| collecting crops | handling the plough | preserving beautiful moments | unloading |
| collecting soil samples | harvesting | preserving natural resources | utilizing riverside path |
| colliding | heading | protecting waterways | vacationing in resort |
| communicating | heading towards | protesters | viewing |
| competing | helping | providing aid | visiting |
| conserving marine environment | hiking | racers | visiting historical landmarks |
| constructing | hindering | racing in adventure challenge | waiting at light |
| contiguous | homeowner and damaged property | racing through city streets | walking |
| cooperating | hosting lakeside wedding | racing through mountain trails | watching |
| coordinated in lane | hosting outdoor concert | raising awareness and funds | watching wildlife |
| coordinating | in conflict | rear | within parking area |
| coordinating logistics | inline | reconstructing | within traffic zone |
| crashing | in the middle | reenacting | worker and construction tools |
| crossing | independent | regulated movement | workers |
| cyclists | inside construction zone | relaxing | working |
| debating | inspecting | repairing | worshipers |
| demonstrating | installing | resident and mudslide debris | serving food |
| directing along | interacting | residents evacuating together | pitching |
| directing toward | investigating | responding | hosting rest stop |
| discussing | isolated during flood | resting | |
| displaying | kicking | retrieving | |
| distributing | lead | revamping urban area | |

## A.2 Annotation Pipeline

To capture frequent and rapid changes in aerial videos while reducing redundancy, we annotate keyframes at 5FPS. At each frame, our annotation pipeline consists of two stages:

**Stage 1: Object Localization and Tracking.** We manually annotate bounding boxes with predefined categories, providing precise object localization and consistent tracking throughout the video.

**Stage 2: Relationship Instance Annotation.** To generate diverse predicates, we leverage the GPT4RoI [70] model, which combines visual and linguistic data to generate detailed descriptions of object relationships within specified regions of interest. The process involves the following steps:

Table A.10: A hierarchical representation of the 249 relationship predicates on the AeroEye dataset, organized into 29 high-level semantic scene categories.

| Scene | Relationship Predictes |
|---|---|
| Baseball | cheering, watching, players, competing, passing, scoring |
| Basketball | cheering, watching, players, competing, dribbling, passing |
| Boating | floating, boaters, navigating, enjoying waterfront gathering |
| Campus | studying, engaging in school sports, learning, graduating |
| Car Racing | racing through city streets, drivers, overtaking, moving |
| Concert | enjoying concert atmosphere, cheering, watching, participating in community event |
| Conflict | in conflict, debating, trying to mediate resolutions, responding |
| Constructing | building urban infrastructure, inside construction zone, worker and construction tools, workers, working, inspecting, installing |
| Cycling | moving, traveling, riding, cyclists, navigating, along the road |
| Fire | extinguishing, responding, firefighters, isolated during flood |
| Flood | rescuing, responding, isolated during flood, floating |
| Harbour | loading cargo onto ship, operating in docking area, floating, navigating |
| Harvesting | collecting crops, farmers, harvesting, tending to agricultural tasks |
| Landslide | rescuing, responding, homeowner and damaged property, isolated during flood, managing industrial operations |
| Mudslide | rescuing, responding, homeowner and damaged property, undergoing process, managing industrial operations |
| NonEvent | isolated during flood, observing, under surveillance |
| Parade Protest | demonstrating, sharing slogans, in line, participating in community event |
| Park | picnicking together, enjoying lakeside park, engaging in leisure activity, walking |
| Party | celebrating, socializing, enjoying outdoor festivities, dancing, singing |
| Ploughing | tending to agricultural tasks, farmers, moving along the lane |
| Police Chase | chasing, responding, following behind, in conflict, moving at intersection |
| Post Earthquake | rescuing, responding, reconstructing, homeowner and damaged property |
| Religious Activity | worshipers, engaging in leisure activity, participating in community event |
| Running | moving, traveling, navigating, runners, participating in marathon |
| Soccer | cheering, watching, players, competing, passing, scoring, kicking, marking |
| Swimming | enjoying lakeside park, enjoying waterfront gathering, swimmers, floating, tackling |
| Traffic Collision | approaching, crashing, responding, colliding, drivers, in the middle of intersection |
| Traffic Congestion | approaching, drivers, waiting at light, along the road, sharing lane |
| Traffic Monitoring | approaching, monitoring, managing urban traffic, observing, under surveillance |

1. **Text Generation:** We leverage GPT4RoI integrateed instruction tuning with a large language model (LLM) to enhance interactions with regions of interest (RoI) within images. This model transforms bounding box references into language instructions, enabling detailed descriptions and reasoning about specific image regions, thus improving image understanding granularity and accuracy. It utilizes a variety of transformed multimodal datasets, including COCO [75] and Visual Genome [23], to refine the alignment between visual and linguistic data, ensuring precise responses to spatial queries.

   In particular, We input bounding boxes around objects with prompts such as, "What is the relationship between `<object_1>` in `<region_1>` and `<object_2>` in `<region_2>`?", where `<object_i>` is a category name that labeled in Stage 1, and GPT4RoI replaces `<region_i>` tags in these instructions with results from RoIAlign, derived directly from the features of image. The model uses RoIAlign to extract region-specific features and combine them with language embeddings. The resulting multimodal embeddings are then interpreted by the Vicuna model [76], an instance of LLaMA [77].

2. **Predicate Summarization and Selection:** We employ a custom-designed filter to categorize the generated text into relationship types. This filter utilizes a combination of keyword matching, dependency parsing, and semantic analysis to identify and classify the predicates accurately. The filter is designed to handle sentence structure and terminology variations, ensuring that the identified predicates are correctly mapped to their corresponding relationship types. Furthermore, the filter incorporates a confidence scoring mechanism to prioritize high-quality predicates and filter out irrelevant or ambiguous ones. The final selection of predicates undergoes human oversight, where experienced annotators review and validate the filtered results. This manual verification step ensures the highest accuracy and relevance of the identified predicates, mitigating potential errors introduced by automated processing.

**Quality Control** To maintain the highest standard of annotation quality, we implement the following comprehensive measures:

- Stage 1: Object Localization and Tracking:

  – All bounding box annotations are performed manually by skilled annotators without relying on automated detection or tracking models. This ensures precise object localization tailored to the specific characteristics of aerial videos.

- We employ a rigorous double-checking process, where each frame in every video is carefully reviewed by a second annotator. This step helps identify and rectify any inaccuracies in bounding box placement or dimensions.
- In cases where object identities are inconsistent across frames due to occlusion, visual similarity, or other challenges, annotators meticulously correct the object numbers to maintain consistent tracking throughout the video.

- Stage 2: Relationship Instance Annotation:
  - Annotators undergo extensive training using carefully curated examples from previous VidSGG datasets. This training familiarizes them with the intricacies of extracting predicates generated by the LLM and ensures a deep understanding of the annotation guidelines and best practices.
  - To minimize individual biases and ensure the robustness of annotations, we implement a repeated annotation process. Each video is distributed to multiple annotators, who independently extract and record the relationship instances. This redundancy allows for cross-validation and helps identify potential discrepancies or ambiguities.
  - In cases where annotators disagree on the extracted predicates or their categorization, a highly experienced meta-annotator is assigned to review the conflicting annotations. The meta-annotator carefully examines the video content, considers the perspectives of the individual annotators, and makes the final decision on the annotation record. This hierarchical review process ensures consistency and accuracy across the dataset.

By employing these rigorous quality control measures at each stage of the annotation pipeline, we ensure the highest level of *accuracy*, *consistency*, and *completeness* in relationship instances.

## A.3 Data Format

Our annotations are stored in JSON (JavaScript Object Notation) format organized as below:

```
data[{
  "file_name": str,
  "height": int,
  "width": int,
  "image_id": int,
  "frame_index": int,
  "video_id": int,
  "metadata":[{
    "id": int,
    "category_id": int,
    "iscrowd": 0 or 1,
    "area": int
  }],
  "annotations":[{
    "bbox": [x, y, width, height],
    "bbox_mode": 0 or 1,
    "category_id": int,
    "track_id": int
  }],
  "positions": [[
    metadata_id,
    metadata_id,
    position_id
  }],
  "relations": [[
    metadata_id,
    metadata_id,
    relation_id
  ]]
}],
"categories": {
```

```
32      "id": int,
33      "name": str
34  },
35  "predicate_positions": {
36      "id": int,
37      "name": str
38  },
39  "predicate_relations": {
40      "id": int,
41      "name": str
42  }
```

**Basic Image Information.** This section contains the fundamental attributes of each image:

- `file_name` (`str`): The name of the image file.
- `height` (`int`): The height of the image in pixels.
- `width` (`int`): The width of the image in pixels.
- `image_id` (`int`): A unique identifier for the image.
- `frame_index` (`int`): The index of the frame within the video sequence.
- `video_id` (`int`): An identifier for the video to which this image/frame belongs.

**Metadata.** This section includes the `metadata` key, which is a list of segments within the image. Each segment contains:

- `id` (`int`): Unique identifier for the segment.
- `category_id` (`int`): Identifier for the category of the object in the segment.
- `iscrowd` (0 or 1): 0 for a single object and 1 for a cluster of objects.
- `area` (`int`): The area covered by the segment in the image.

The `annotations` key contains a list of corresponding bounding boxes for each entry in `metadata`, each tagged with a specific `category_id`:

- `bbox` (`list`): [`x_center, y_center, width, height`] of the bounding box.
- `bbox_mode` (0 or 1): Bounding box mode.
- `category_id` (`int`): Identifier for the object category in the bounding box.
- `track_id` (`int`): Identifier to track the bounding box across different frames.

**Relationship Attributes.** This section encompasses lists of `positions` and `relations` for each segment, including two different `metadata_ids` to represent the interactivity between two segments:

- `positions` (`list`): List of position relations between segments, each containing:
    - `metadata_id` (`int`): Identifier for the first segment.
    - `metadata_id` (`int`): Identifier for the second segment.
    - `position_id` (`str`): Identifier for position relation between the segments.
- `relations` (`list`): List of other relations between segments, each containing:
    - `metadata_id` (`int`): Identifier for the first segment.
    - `metadata_id` (`int`): Identifier for the second segment.
    - `relation_id` (`str`): Identifier for relationship between the segments.

These descriptors represent lists specifying various subject, object, and interactivity aspects for each bounding box within the annotations and `metadata`. For example, [3, 0, 5] indicates that the third and first metadata segments share the relationship with an ID of 5.

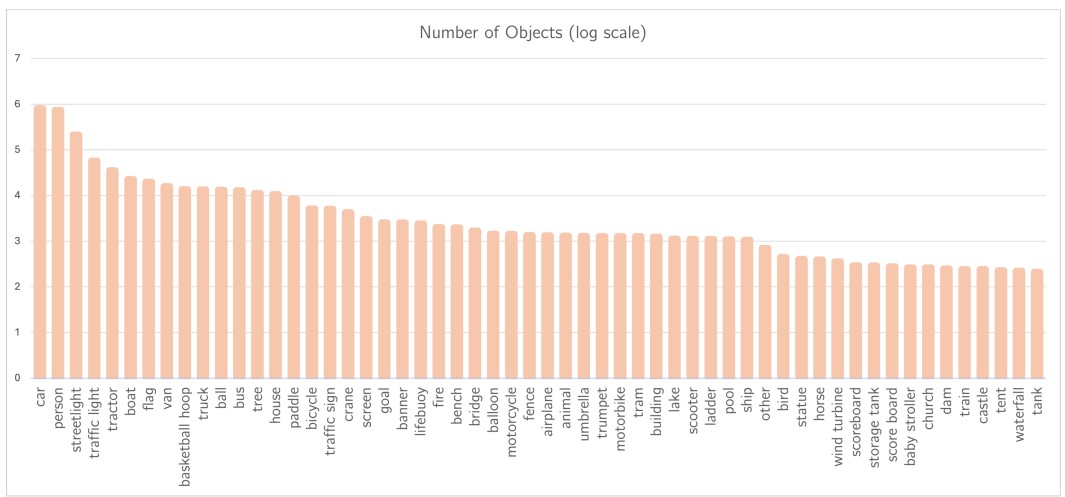

Figure A.8: Distribution of objects per category on the AeroEye dataset.

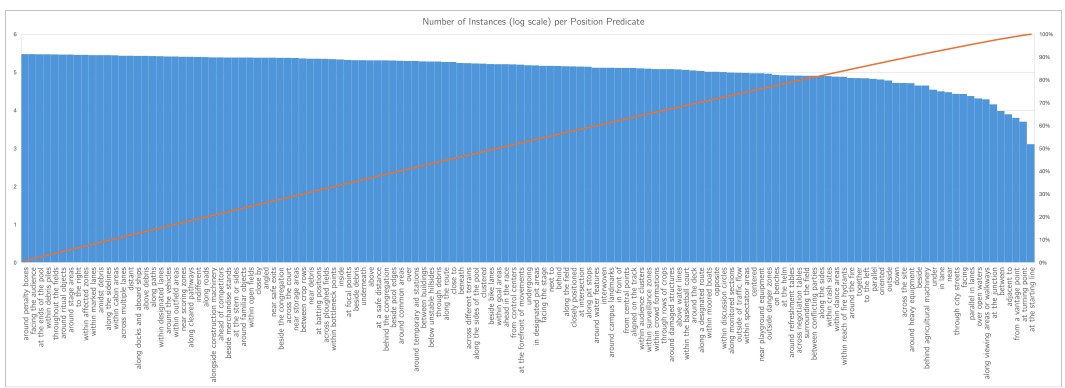

Figure A.9: Distribution of position predicates per category on the AeroEye dataset.

## A.4 Additional Statistics

We present object, position, and relationship statistics per category in Fig. A.8, A.9, and A.10.

## A.5 Data Samples

Fig. A.11 presents selected samples from our *AeroEye* dataset, distinguished by its detailed bounding box annotations and meticulous relationship descriptions across various scenarios. As outlined in Section 3.2, each frame within AeroEye is annotated with precision and contextual relevance, ensuring clarity and avoiding the common ambiguities, such as generic or overlapping labels, prevalent in other datasets. A key feature of AeroEye is its categorization of relationships into positions and relations, as illustrated by the curved arrows representing positions and straight arrows denoting relations in Fig. A.11. This multifaceted approach to annotation renders AeroEye uniquely comprehensive when compared to other datasets [6, 27, 7, 8]. The meticulous annotation process and the meticulously curated, information-rich nature of the AeroEye dataset firmly establish it as an invaluable resource, poised to catalyze significant advancements in relationship modeling within drone videos.

# B Concepts

## B.1 Progression

**Relationship Representation.** In each frame, a detector provides to a set of object features $\{\widehat{v}_t^1, \ldots, \widehat{v}_t^{N(t)}\}$, bounding boxes $\{\widehat{b}_t^1, \ldots, \widehat{b}_t^{N(t)}\}$, and category distributions $\{\widehat{d}_t^1, \ldots, \widehat{d}_t^{N(t)}\}$ for

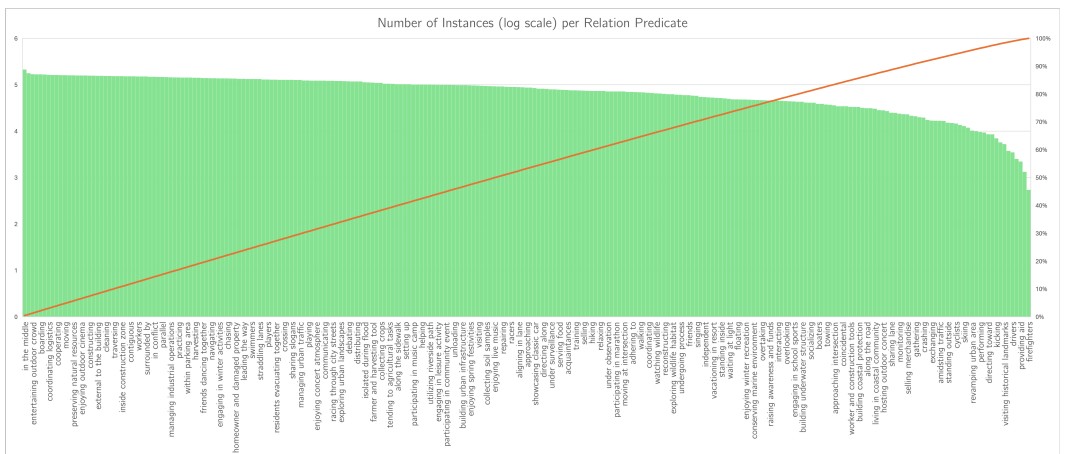

Figure A.10: Distribution of relationship predicates per category on the AeroEye dataset.

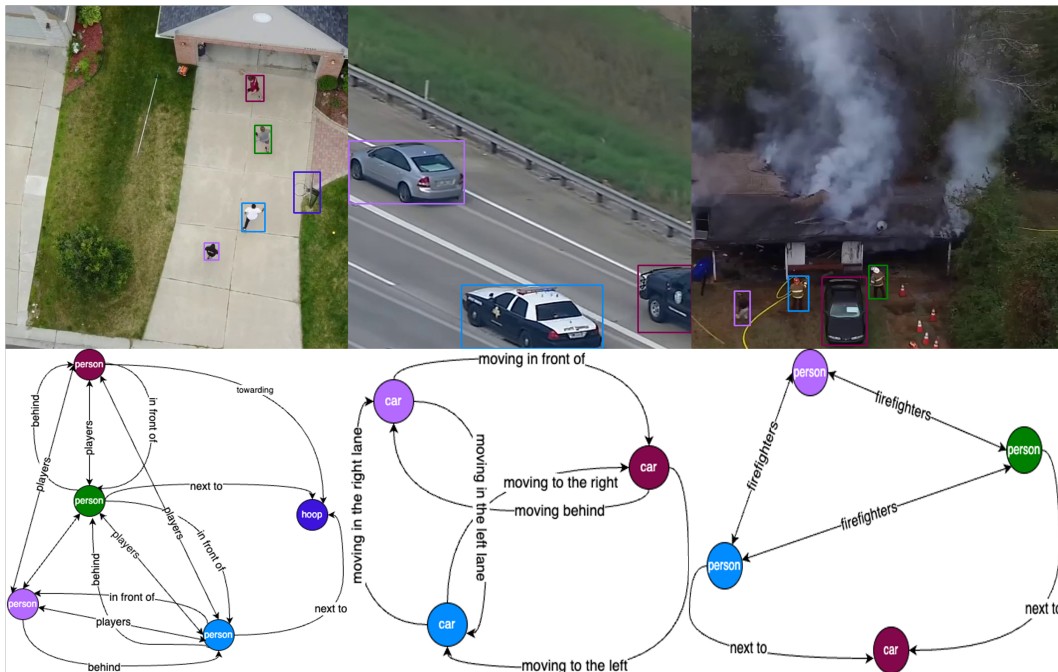

Figure A.11: Our *AeroEye* dataset includes a diverse range of scenarios, objects, and relationships.

the detected objects. These elements are used to define the relationships $\widehat{x}_t^k$:

$$\widehat{x}_t^k = \left[ \phi_{W_s}\widehat{z}_i^t, \phi_{W_o}\widehat{z}_j^t, \phi_{W_u}\varphi(\widehat{u}_t^{ij} \odot f(\widehat{b}_t^i, \widehat{b}_t^j)), \widehat{s}_t^i, \widehat{s}_t^j \right] \tag{B.7}$$

where $[\cdot, \cdot]$ denotes concatenation, $\varphi$ denotes a flattening operation, $\odot$ represents element-wise addition, and the linear transformation matrices $\phi_{W_s}$, $\phi_{W_o}$, and $\phi_{W_u}$. $\widehat{u}_{ij}^t$ implies the feature map of the union box, computed by the RoIAlign [78], while $f$ is a function that converts the bounding boxes of the subject $\widehat{b}_i^t$ and object $\widehat{b}_j^t$ into a feature map with the same dimensions as $\widehat{u}_{ij}^t$. The semantic embedding vectors $\widehat{s}_i^t$ and $\widehat{s}_j^t$ are obtained from the object categories $\widehat{c}_i^t$ and $\widehat{c}_j^t$, respectively.

In the progressive approach, features $v_t^i$ for each object are obtained using a detector such as Faster R-CNN in each frame. These features are utilized as specified in Eqn. (B.7) to compute the relationship features $\widehat{x}_t^k$. Finally, these features are processed through multilayer perceptions (MLPs) followed by softmax activation functions to classify the types of relationships between objects.

## B.2 Batch Progression

In contrast to the Progressive approach, the Batch Progressive approach passes the relationship features through a Transformer architecture before classification.

A set of relationship features $\widehat{X}_t = \{\widehat{x}_t^1, \widehat{x}_t^2, \ldots, \widehat{x}_t^{K(t)}\}$ is fed into the Transformer **_Encoder_**, which focuses on understanding the spatial context by inputting these relationship features into a sequence of identical self-attention layers, defined as:

$$\widehat{X}_t^{(n)} = \text{Att}_{\text{enc.}}(Q = K = V = \widehat{X}_t^{(n-1)}) \tag{B.8}$$

Each $n$-th layer receives the output from the $(n-1)$-th layer as its input, iteratively refining to enhance the representation of spatial relations embedded in the features, where $n$ denotes the layer number. The outputs are then processed by the **_Decoder_**, which captures temporal dependencies between frames, applying a sliding window over the sequence of spatially contextualized representations:

$$\widehat{Z}_i = [\widehat{X}_i, \ldots, \widehat{X}_{T-\eta-1}], \quad i \in \{1, \ldots, T\} \tag{B.9}$$

where $\eta$ is the window size, and $T$ is number of frames. The positional encoding $E_f = [e_1, \ldots, e_\eta]$ is embedded into the input to maintain sequence order:

$$Q = K = \widehat{Z}_i + E_f, V = \widehat{Z}_i, Z_i = \text{Att}_{\text{dec.}}(Q, K, V). \tag{B.10}$$

Equation (B.10) enables the decoder to process each batch using self-attention layers, combining relation representations $\widehat{Z}_i$ with positional encodings $E_f$.

## B.3 Hierarchical Graph

The hierarchical interlacement graph (HIG) method abstracts video content by representing temporal relationships using a multi-level graph structure. Higher-level graph cells encompass broader segments of video frames, allowing the HIG to efficiently capture and model the temporal dependencies and connections across different time scales within the video. In this method, object features $\widehat{v}_t^i$ from individual frames are spatially and temporally fused to form graph nodes. These nodes are interconnected across successive frames, resulting in a series of interconnected frame-based graphs $\{G_t(V_t, E_t)\}_{t=1}^T$, where each $G_t$ is defined by its vertices $V_t$ and edges $E_t$. As the graph traverses, the total number of graphs decreases, ultimately resulting in a singular graph representing the entire video. This hierarchical graph operates on predefined hierarchical levels $L$. At each level $l$, the temporal scope is adjusted to $T_l = T - l + 1$, and a new graph is generated that is specific to that level and timeframe. Therefore, node features within each graph are dynamically updated through the computation and aggregation of messages from adjacent nodes, computed by weight matrices for each level $l$ and node pair $(\widehat{v}_i, \widehat{v}_j)$. This iterative refinement process is applied across all levels, resulting in a consolidated graph structure and updated feature set. Finally, relationship features between each node pair $(\widehat{v}_i, \widehat{v}_j)$ are fused and analyzed to classify the relationships.

