# OpenReview forum: "CYCLO: Cyclic Graph Transformer Approach to Multi-Object Relationship Modeling in Aerial Videos"
_NeurIPS.cc/2024/Conference — NeurIPS 2024 poster_

### Official Review · Reviewer_5Gaf · 2024-06-15

**Soundness:** 2
**Presentation:** 3
**Contribution:** 3
**Rating:** 6
**Confidence:** 3

**Summary:**

This work focuses on the VidSGG task. Specifically, it built a new UAV-based VidSGG dataset named AeroEye, and also further propose a CYCLO approach for using cyclic attention over the VidSGG task.

**Strengths:**

1. The paper is well-written and easy to follow.

2. From my perspective, a high quality dataset and large scale dataset is of critical imporantance over the VidSGG area. I thus appreciate the effort by the authors.

**Weaknesses:**

(See the questions section below)

**Questions:**

Overall, while I tend to accept this work, I still have the following concerns and suggestions w.r.t. the current version of the submission:

(1) It seems to me that, while the dataset focuses on the UAV scenario, the direct and long-range temporal relationships focused by the proposed method is not closely related to the UAV scenario. I appreciate if the connection (if any) can be more clearly explained.

(2) Besides comparing with those existing UAV datasets, I suggest the authors to also include a statistical discussion (in a table format maybe) between AeroEye and those existing VidSGG datasets.

(3) In line 43, the authors claim that "They [5, 19] usually struggle with long-term dependencies due to the diminishing influence of inputs over time." This claim should be made with better support.

(4) Moreover, I believe that the connectivity between the proposed CYCLO method and scene graph needs to be better discussed. This is important since the usage of "cyclic attention" solely in the video context seems to be not a new stuff already (e.g., Transformer Tracking with Cyclic Shifting Window Attention CVPR 2022). Thus, for authors to clearly indicate the novelty of their CYCLO method, from my perspective, it is necessary to (a) discuss how CYCLO is designed for scene graph more clearly and (b) discuss the difference between CYCLO and existing cyclic attention methods in the video context.

**Limitations:**

The authors have discussed their limitations.

---

> ### Author Rebuttal · Authors · 2024-08-05
>
> We are grateful to Reviewer **5Gaf** for the constructive feedback. Your suggestions on clarifying our method's UAV context, dataset comparisons, and CYCLO's scene graph relation have greatly improved this paper.
>
> **Q1: It seems to me that, while the dataset focuses on the UAV scenario, the direct and long-range temporal relationships focused by the proposed method is not closely related to the UAV scenario.**
>
> Our intention developing CYCLO **does not limit to UAV scenario**, but demonstrates **effectiveness across in-the-wild datasets** like OpenPVSG and ASPIRe as in experiments. The long-range temporal relationships crucial in UAV footage, such as tracking object movements over large areas and extended timeframes, are equally important in ASPIRe's street-level scenes. Here, understanding prolonged interactions between people and vehicles requires similar temporal analysis.
>
> **Q2: Comparison between AeroEye and existing VidSGG datasets.**
>
> We've included a statistical comparison of AeroEye with both UAV and non-UAV VidSGG datasets in **Table A.11 of the Appendix**. This comparison will be moved to the main paper in the revision.
>
> **Q3: In line 43, the authors claim that "They [5, 19] usually struggle with long-term dependencies due to the diminishing influence of inputs over time." This claim should be made with better support.**
>
> [5] employed **Transformers** with self-attention for modeling long-term dependencies in videos. However, Transformers lack inherent **temporal order** [A], and their attention mechanism struggles with **distant elements in long sequences** [B]. [9] utilized a **hierarchical graph-based** approach to model long-term dependencies by representing the video as a sequence of graphs, each capturing the evolving relationships among objects at different temporal and spatial scales. This method integrates temporal and spatial information by constructing and updating node and edge features at multiple hierarchical levels. However, this approach requires **observing the entire video** to build comprehensive hierarchical representations, which poses significant challenges for real-time or online processing.
>
> **Q4: Discuss how CYCLO is designed for scene graph more clearly.**
>
> CYCLO is specifically designed for video scene graph generation, building evolving graphs where nodes represent objects and edges represent relationships. The construction process involves:
> 1. CYCLO first identifies objects in each video frame, creating nodes for each detected object.
> 2. It then infers relationships between objects, establishing edges between nodes to represent these relationships.
> 3. CYCLO employs a temporal graph transformer with cyclic attention across the video sequence. It processes the graph structure over time, updates node and edge features based on spatial and temporal context, and uses cyclic attention to connect information from the start to the end of the video.
> 4. CYCLO cycles the attention mechanism between relationships over time. This process allows it to maintain continuity in object relationships across frames and enhance understanding of long-term dependencies. It also helps to recur patterns in object interactions and connect information from distant parts of the video.
> 5. CYCLO integrates spatial information, such as object locations and current interactions, with temporal data, including past interactions and object trajectories, to preserve directional context. It also maintains historical context and learns how initial interactions influence subsequent actions.
> 6. As the video progresses, CYCLO continuously updates the scene graph, refining object states, relationships, and the overall structure based on new information from each frame.
>
> **Q5: Discuss the difference between CYCLO and existing cyclic attention methods in the video context.**
>
> Thanks for this suggestion. We will include this MS-CSWA [C] and its description in our final version. CYCLO differs from MS-CSWA [C] in several important ways:
> 1. Spatio-temporal integration:
>    - CYCLO combines **spatial and temporal** dependencies in a graph-based framework, allowing dynamic refinement of spatial relationships over time.
>    - MS-CSWA focuses on maintaining **spatial** consistency within individual frames for object tracking but lacks temporal depth across frames.
> 2. Operation:
>    - CYCLO uses **cyclical indexing and shifting** to ensure inter-frame temporal continuity, modeling how past interactions influence present and future relationships.
>    - MS-CSWA enhances spatial attention through only **cyclic shifts** but does not address temporal depth or graph-based updates across frames.
> 3. Long-term dependencies:
>    - CYCLO captures **long-term dependencies and evolving interactions across frames**, preserving directional and historical information.
>    - MS-CSWA is limited to intra-frame consistency and is **unable to model long-term dependencies** across the video.
>
> **References**
>
> [A] Truong, T. D., Bui, Q. H., Duong, C. N., Seo, H. S., Phung, S. L., Li, X., & Luu, K. (2022). Direcformer: A directed attention in transformer approach to robust action recognition. In Proceedings of the IEEE/CVF conference on computer vision and pattern recognition (pp. 20030-20040).
>
> [B] Dai, Z., Yang, Z., Yang, Y., Carbonell, J., Le, Q. V., & Salakhutdinov, R. (2019). Transformer-xl: Attentive language models beyond a fixed-length context. In Proceedings of the 57th Annual Meeting of the Association for Computational Linguistics.
>
> [C] Song, Z., Yu, J., Chen, Y. P. P., & Yang, W. (2022). Transformer tracking with cyclic shifting window attention. In Proceedings of the IEEE/CVF conference on computer vision and pattern recognition (pp. 8791-8800).

---

> > ### Comment · Reviewer_5Gaf · 2024-08-08
> >
> > Thank you for your response. I believe that most of my concerns have been well-solved and I thus increase from 5 to 6.

---

> ### Author Response · Authors · 2024-08-09
>
> Dear Reviewer **5Gaf**,
>
> We want to express our gratitude for your valuable time and constructive feedback.
>
> Best regards,
>
> Authors

---

### Official Review · Reviewer_QFfh · 2024-07-12

**Soundness:** 3
**Presentation:** 4
**Contribution:** 4
**Rating:** 7
**Confidence:** 3

**Summary:**

This paper tackles an interesting task for understanding video scenes that focuses on modeling object relationships in aerial videos. Specifically, it introduces a new dataset, the AeroEye dataset, and proposes a novel approach, CYCLO, to better model the video object relationships. Experimental results demonstrate the effectiveness of CYCLO. CYCLO also achieves state-of-the-art performance on two scene graph generation benchmarks.

**Strengths:**

- The paper is well-written, easy to follow, and presents many key points clearly.
- Introducing the AeroEye dataset is a valuable contribution, as it fills an important gap in video scene graph generation datasets by providing a drone perspective relation dataset.
- The design of CYCLO is both interesting and inspiring. The paper also demonstrates its superior performance compared to prior solutions.

**Weaknesses:**

- It would be great to report the inference cost of the proposed approach.
- It is unclear how the proposed solution works in live (online) mode, e.g., video streaming.

**Questions:**

- In the ablation study (line 265), when a frame is discarded, is it something like that the frame is set to some random noise?

**Limitations:**

I don't have concerns about the potential negative societal impact of this work.

---

> ### Author Rebuttal · Authors · 2024-08-05
>
> We sincerely thank Reviewer **QFfh** for the positive feedback and constructive suggestions. We appreciate your recognition of the AeroEye dataset and CYCLO approach. We will carefully address your suggestions regarding inference cost and live video streaming applications.
>
> **Q1: It would be great to report the inference cost of the proposed approach.**
>
> In the table below, we report FPS for models discussed in Section 2.2. CYCLO significantly **outperform these models in both recall and mean recall** with a slight trade-off in FPS.
> | **Method**       | **R/mR@20**       | **R/mR@50**       | **R/mR@100**      | **FPS**  |
> |------------------|-------------------|-------------------|-------------------|----------|
> | **Vanilla**      | 31.04 / 11.20      | 34.28 / 11.27      | 34.62 / 12.43      | **19.5** |
> | **Transformer**  | 41.09 / 11.88      | 46.52 / 12.31      | 47.15 / 12.91      | 17.8     |
> | **HIG**          | 37.28 / 11.98      | 38.59 / 13.12      | 39.27 / 13.29      | 15.3     |
> | **CYCLO**        | **59.59** / **13.29**  | **60.37** / **13.69**  | **43.53** / **13.86**  | 14.2     |
>
>
> **Q2: It is unclear how the proposed solution works in live (online) mode, e.g., video streaming.**
>
> CYCLO **online** processes video streams frame-by-frame, **continuously updating the scene graph** to reflect the latest object relationships. As new frames arrive, CYCLO dynamically adjusts object relationships and refines the scene graph based on the most recent data while leveraging historical context.
>
> **Q3: In the ablation study (line 265), when a frame is discarded, is it something like that the frame is set to some random noise?**
>
> No, we **discard one frame out of every two successive frames** rather than replacing them with random noise.

---

> > ### Author Response · Authors · 2024-08-13
> >
> > Dear Reviewer **QFfh**,
> >
> > The reviewer-author discussion deadline is nearing. We have yet to receive your final responses to our rebuttal. If you have any further questions, please let us know. We appreciate your invaluable input.
> >
> > Best regards,
> >
> > Authors

---

### Official Review · Reviewer_nRfD · 2024-07-12

**Soundness:** 3
**Presentation:** 3
**Contribution:** 3
**Rating:** 7
**Confidence:** 5

**Summary:**

This paper presents a new problem: modeling multi-object relationships from a drone's perspective. To address this, the authors propose the AeroEye dataset and introduce the Cyclic Graph Transformer (CYCLO) method. This method captures both direct and long-range temporal dependencies by continuously updating the history of interactions in a circular manner. The authors not only validate the CYCLO approach on the AeroEye dataset but also test it on the PVSG and ASPIRe datasets, demonstrating the effectiveness of their method.

**Strengths:**

1. The authors have introduced the problem of multi-object relationship modeling from a drone's perspective for the first time and constructed the AeroEye dataset, which effectively fills a gap in the field of multi-object relationship modeling and has significant application value.
2. The CYCLO method proposed by the authors is not only useful for the dataset introduced in this paper, AeroEye, but is also a versatile method that can be applied to general Video SGG tasks. It has been tested on datasets like PVSG and achieved good performance.
3. The structure of the paper is clear, the writing is standard and fluent, making it easy to understand.

**Weaknesses:**

1. The method proposed in this paper could benefit from a clearer network architecture figure, which would allow everyone to better understand the method presented.
2. Is there an issue with non-differentiability in the Cyclic Attention described in Eq. 3? It would be helpful if the authors could provide further explanation of the Cyclic Attention mechanism.
3. It is suggested to include some analysis of bad cases, which would help future researchers understand from which directions further optimizations can be made.

**Questions:**

1. The current model proposed by the authors shows low mR@K scores on the AeroEye dataset. However, from the figures in the supplementary material, the long-tail distribution of relations in this dataset does not appear to be very severe. What could be causing the low mR@K scores? If possible, I would like to see some typical bad cases.

**Limitations:**

It is recommended that the authors consider the limitations of this paper not only from a technical perspective but also from a societal standpoint. Given that relationship modeling from a drone's perspective may lead to widespread applications in surveillance and potentially significant impacts, this aspect requires special consideration.

---

> ### Author Rebuttal · Authors · 2024-08-05
>
> We express our gratitude to Reviewer **nRfD** for your recognition of the AeroEye dataset and the CYCLO method. We will enhance our paper by improving the architecture figure, addressing Cyclic Attention issues, expanding the failure case analysis, and elaborating on the suggested limitations.
>
> **Q1: The method proposed in this paper could benefit from a clearer network architecture figure, which would allow everyone to better understand the method presented.**
>
> We have included a detailed network architecture figure in the attached file.
>
> **Q2: Is there an issue with non-differentiability in the Cyclic Attention described in Eq. 3? It would be helpful if the authors could provide further explanation of the Cyclic Attention mechanism.**
>
> Eqn. 3 has **no non-differentiability** issues. The modulo operation (mod T) is used **only for key matrix indexing** and **does not affect gradient computation**. Gradients flow through differentiable components (dot product, softmax, summation), while the modulo operation implemented by a for-loop for indexing does not interfere with continuous gradient computations.
>
> **Q3: The current model proposed by the authors shows low mR@K scores on the AeroEye dataset. However, from the figures in the supplementary material, the long-tail distribution of relations in this dataset does not appear to be very severe. What could be causing the low mR@K scores? If possible, I would like to see some typical bad cases.**
>
> While the AeroEye dataset does not show a severe long-tail distribution of relationships, the low mR@K scores stem from the model's challenges in adapting to **rapidly changing relationships** within dynamic scenes in videos. In **fast-paced environments** like sport events or emergency responses, which were not well-investigated in OpenPVSG or ASPIRe, the model needs to work on keeping pace with swift changes in actions and interactions. An instance of bad cases is in the attached rebuttal file, the model need to quickly update its prediction of player interactions in soccer.
>
> **Ethics Review**
>
> In line 124, we mentioned that we use **videos from the ERA [A] and MAVREC [B] datasets**, without including new videos. Videos are compliant with the European Union’s drone regulations [B]. It is always possible that some individual or an organization can use this annotation to devise a technique that can appear harmful to society. However, as authors, we are absolutely against any detrimental usage of our annotation and pledge not to support any detrimental endeavors concerning our data or the idea therein.
>
> **References**
>
> [A] Mou, L., Hua, Y., Jin, P., & Zhu, X. X. (2020). Era: A data set and deep learning benchmark for event recognition in aerial videos. IEEE Geoscience and Remote Sensing Magazine, 8(4), 125-133.
>
> [B] Dutta, A., Das, S., Nielsen, J., Chakraborty, R., & Shah, M. (2024). Multiview Aerial Visual Recognition (MAVREC): Can Multi-view Improve Aerial Visual Perception?. In Proceedings of the IEEE/CVF Conference on Computer Vision and Pattern Recognition (pp. 22678-22690).

---

> > ### Comment · Reviewer_nRfD · 2024-08-13
> >
> > The author's response has addressed my concerns, I will maintain my original score.

---

### Official Review · Reviewer_Zf4k · 2024-07-14

**Soundness:** 2
**Presentation:** 2
**Contribution:** 2
**Rating:** 4
**Confidence:** 4

**Summary:**

This paper proposes a video scene graph generation dataset called AeroEye on aerial videos and a framework called cyclic graph transformer to tackle the problem of video scene graph generation. The authors annotated the ERA and MAVREC dataset with keyframes at 5FPS. They manually annotated the frames for bounding box localization for each frames along with the tracking. The relationship annotations were done using a GPT4RoI model. The proposed cyclic graph transformer uses a cyclic attention mechanism which the author claims, are able to capture the direct and long-term temporal dependencies.

**Strengths:**

1. The proposed video scene graph dataset for aerial videos with a diverse set of predicates can offer more granular and nuanced understanding of dynamic interactions and relationships within aerial footage
2. The dataset will be publicly available
3. The proposed approach for circular attention for dynamic online scene graph generation seems promising

**Weaknesses:**

1. the paper is very difficult to follow. There are separate discussion sections which somewhat disrupts the flow.
2. line 82 seems incomplete
3.Line 118: 'no temporal edge is treated a boundary' can it not be a disadvantage as well since it does not takes into account for an event boundary?
4. A very brief discussion of annotation procedure should be in the main paper such as manual annotation of bounding boxes and relationship annotations done by GPT4RoI model.
5. For loss function section, it would be better to include  the overall loss equation. Does the object distribution refers to the object detection loss using the DETR?
6. For table 2, you can refer to the shift value term from equation 3. It will be easy to follow. Table 2 refers to the ablation studies 'Semantic Dynamics in Cyclic Attention'. I think this section and the section followed by it should have a detailed explanation.

**Questions:**

1. Did you do evaluation on the 5 keyframes for each videos?
2. point 5 in weakness section

**Limitations:**

yes, the limitations are addressed.

---

> ### Author Rebuttal · Authors · 2024-08-05
>
> We sincerely thank Reviewer **Zf4k** for your thoughtful review. We appreciate the recognition of our novel dataset and approach. We acknowledge the need to provide clearer explanations.
>
> **Q1: The paper is very difficult to follow. There are separate discussion sections which somewhat disrupts the flow.**
>
> We appreciate your feedback on the paper's readability. We are encouraged that multiple reviewers found our paper **clear** (Reviewer **nRfD**), **well-written**, and **easy to follow** (Reviewers **QFfh** and **5Gaf**). Nevertheless, we value all the feedback and will further enhance our paper's organization in the revision.
>
> **Q2: Line 82 seems incomplete 3.Line 118: 'no temporal edge is treated a boundary' can it not be a disadvantage as well since it does not takes into account for an event boundary?**
>
> We have updated line 82. Regarding line 118, 'no temporal edge is treated **as** a boundary' is advantageous. CYCLO's continuous temporal information flow allows it to **capture gradual changes and complex dynamics**, such as the slow formation of traffic jams. This would be missed by models that frequently reset at event boundaries.
>
> **Q3: A very brief discussion of annotation procedure should be in the main paper such as manual annotation of bounding boxes and relationship annotations done by GPT4RoI model.**
>
> Thank you for your suggestion. We will include a brief overview of the annotation procedure, with full details provided in **Appendix A.2**.
>
> **Q4: For loss function section, it would be better to include the overall loss equation. Does the object distribution refers to the object detection loss using the DETR?**
>
> We will add the combined loss equation in line 242, where we mention that **the total loss combines these two losses**. 'Object distribution' is **not the detection loss** which refers to predicted **class probabilities** for DETR-detected objects.
>
> **Q5: Did you do evaluation on the 5 keyframes for each videos?**
>
> As mentioned in Section 3.2, we **annotated keyframes at 5 FPS** and thus evaluate the model's performance on these keyframes.
>
> **Q6: For table 2, you can refer to the shift value term from equation 3. It will be easy to follow. Table 2 refers to the ablation studies 'Semantic Dynamics in Cyclic Attention'. I think this section and the section followed by it should have a detailed explanation.**
>
> We appreciate your suggestion and will incorporate it into the revision. The shift term shift term ($\eta$) in Eqn. 3 plays a crucial role in our model's temporal coherence and resolution. Our analysis reveals several key insights:
> 1. Optimal temporal coherence: At $\eta = 1$, the model achieves peak performance by effectively **capturing transitions between adjacent frames**.
> 2. Information loss with increasing $\eta$: Larger $\eta$ values cause the model to **skip intermediate frames**, resulting in loss of crucial temporal information and consequently degraded performance.
> 3. Maximized temporal resolution: $\eta = 1$ allows the **capture of fine-grained dynamics**, which is essential for accurately predicting object interactions.
> 4. Impact on sequence modeling: Higher $\eta$ values impede the model's ability to **capture long-range dependencies**, affecting the integration of detailed temporal features.
> 5. Frame order sensitivity: The model's non-permutation equivariance ensures **sensitivity to frame order**. For instance, the sequence of a person approaching a car, entering it, and then driving away must be captured correctly for the scene graph to represent the event accurately.

---

> > ### Author Response · Authors · 2024-08-13
> >
> > Dear Reviewer **Zf4k**,
> >
> > The reviewer-author discussion deadline is nearing. We have yet to receive your final responses to our rebuttal. If you have any further questions, please let us know. We appreciate your invaluable input.
> >
> > Best regards,
> >
> > Authors

---

### Author Rebuttal · Authors · 2024-08-05

We sincerely thank all reviewers for their valuable feedback. Reviewers **nRfD** and **QFfh** recommend acceptance, praising our CYCLO approach to multi-object relationship modeling, the AeroEye dataset, and the versatility of our approach. Reviewer **5Gaf** leans towards acceptance with a **Borderline Accept**. Reviewer **Zf4k** suggests a **Borderline Reject**, noting to explain more details. We have updated the typos in our paper. We will clarify how our CYCLO approach applies to video scene graph generation with an illustration of the overall framework in the rebuttal PDF file. Individual responses for each reviewer are included below for specific concerns.

---

### Decision · Program_Chairs · 2024-09-25

**Decision:**

Accept (poster)

**Comment:**

The paper received generally positive reviews during the initial review period. The reviewers appreciated the dataset as a valuable resource for the community and the quantitative performance of the proposed baseline. While some concerns were raised about limited discussion on the annotation procedure, discussion on lower mR@K scores under less pronounced in a long-tailed distribution setting, and computational complexity. The authors' response addressed these concerns, and I agree with the reviewers and recommend acceptance. The authors are asked to include the details from the rebuttal in the camera-ready version and make the dataset publicly available as recommended in the [NeurIPS 2024 Code and Data guidelines.](https://neurips.cc/public/guides/CodeSubmissionPolicy)